# LEARNING VIDEO REPRESENTATIONS WITHOUT NATURAL VIDEOS

## ABSTRACT

In this paper, we show that useful video representations can be learned from synthetic videos and natural images, without incorporating natural videos in the training. We propose a progression of video datasets synthesized by simple generative processes, that model a growing set of natural video properties (e.g. motion, acceleration, and shape transformations). The downstream performance of video models pre-trained on these generated datasets gradually increases with the dataset progression. A VideoMAE model pre-trained on our synthetic videos closes 97.2% of the performance gap on UCF101 action classification between training from scratch and self-supervised pre-training from natural videos, and outperforms the pre-trained model on HMDB51. Introducing crops of static images to the pre-training stage results in similar performance to UCF101 pre-training and outperforms the UCF101 pre-trained model on 11 out of 14 out-of-distribution datasets of UCF101-P. Analyzing the low-level properties of the datasets, we identify correlations between frame diversity, frame similarity to natural data, and downstream performance. Our approach provides a more controllable and transparent alternative to video data curation processes for pre-training[1].

## 1 INTRODUCTION

Large-scale data is a fundamental component for training neural networks in various domains, such as natural language processing (NLP). To learn from such data, a prevalent technique is to pre-train models via a self-supervised task (e.g. masked modeling (Devlin et al., 2019) or next-token prediction (Radford et al., 2018; Brown et al., 2020). Adapting these models to downstream tasks results usually in improvements in various NLP tasks.

While self-supervised pre-training is successful in NLP, the same level of success has not been achieved yet in computer vision. Specifically, in the video domain, although various large-scale datasets exist and have been incorporated via similar self-supervised learning tasks, the improvements in downstream performance on video understanding (e.g. action recognition) are relatively low.

One hypothesis for the limited success of self-supervised learning from videos is that current methods fail to effectively utilize the natural video data and learn useful video representations from it. To investigate this hypothesis, we ask if natural videos are even needed to learn video representations that are similar in performance to current state-of-the-art representations.

In this work, we reach a downstream performance that is similar to the performance of models pre-trained on natural videos, while pre-training solely on simple synthetic videos and static images. We propose a progression of simple synthetic video generators that model a *gradually growing set of video data properties* - starting from static frames with solid-color circles and introducing additional shapes, dynamics, temporal shape changes, acceleration, and other textures). We show that adding each of the different properties improves the downstream video understanding performance.

Surprisingly, we find that the gap between the performance of our models and models that were pre-trained on natural videos is minor when we pre-train using purely synthetic data, and eliminated when we introduce natural image crops. By pre-training a VideoMAE (Wang et al., 2023) on purely generated data we close 97.2% of the gap in UCF101 classification accuracy between a model that

---

[1]Code, datasets, and models will be provided upon acceptance.

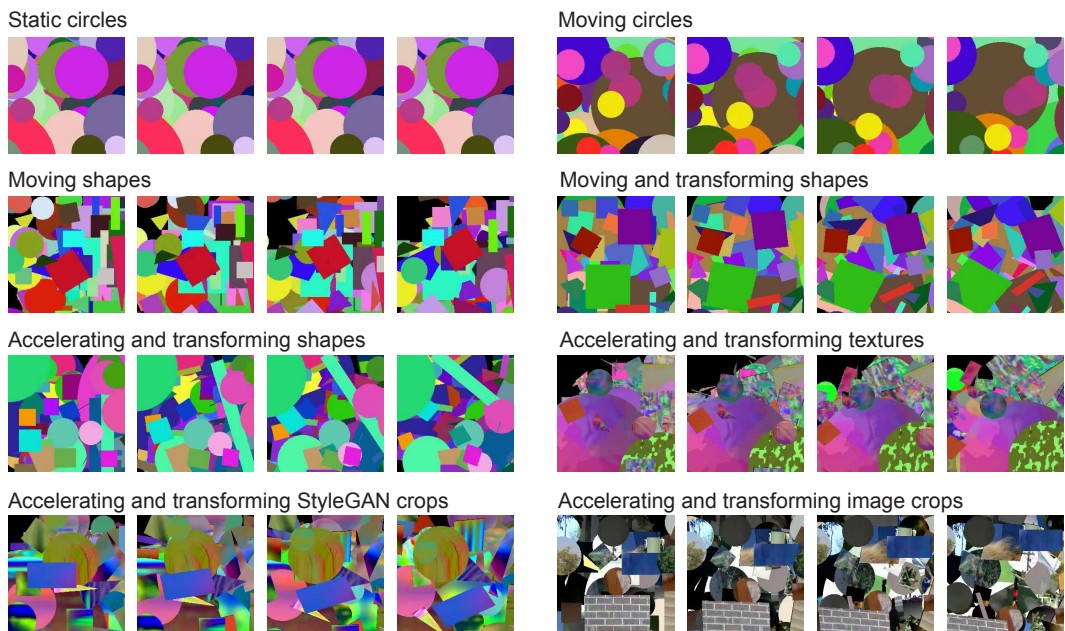

Figure 1: **Samples from our progression of video generation models and additionally included image datasets.** We present 4 frames from timestamps $t \in \{0, 10, 20, 30\}$ of a randomly sampled video from each of our generated datasets, and UCF101 (left to right).

was trained from scratch and a model that was pre-trained on UCF101. By incorporating additional crops from *static images*, the performance of our models matches or improves upon the performance of the UCF101 pre-trained model.

When evaluating performance on an out-of-distribution dataset, UCF101-P (Schiappa et al., 2023), the last models in our progression perform *better* than a model that was pre-trained on UCF101 in 11 out of 14 corrupted dataset versions. This shows the additional benefit of training on synthetic data, and that representations of current state-of-the-art models are less reliable in out-of-distribution settings than our alternative approach, for which is generation process is fully transparent.

Finally, by comparing the accuracy of models pre-trained on the generated data in the progression, we identify different data properties that correspond with improved downstream performance. Specifically, we find that high velocities and accelerations of moving shapes in the video, as well as similarity in the color space to natural videos and high frame diversity, correlate to high action recognition accuracy. We believe that these observations can help to guide future practices for large-scale self-supervised video learning.

## 2 RELATED WORK

**Video representation learning.** Learning useful representation for videos is a widely explored problem. Early methods used models that were pre-trained on image datasets and fine-tuned them on videos (Simonyan & Zisserman, 2014; Tran et al., 2018).

Following the success of self-supervised representation learning (SSL) for images, similar approaches were applied to learn from videos. Earlier SSL approaches designed pretext tasks that rely on known video properties (e.g. temporal smoothness) - classifying videos to ordered or shuffled frames (Misra et al., 2016; Xu et al., 2019), predicting the future frames (Mathieu et al., 2015), predicting the arrow of time (Wei et al., 2018), and predicting the speed of the video (Benaim et al., 2020).

Recently, VideoMAE (Tong et al., 2022), MAE-ST (Feichtenhofer et al., 2022), and VideoMAE-V2 (Wang et al., 2023) used variations of masked auto-encoding (He et al., 2022) and trained a transformer to predict masked temporal video patches as the pretext task. These approaches were

shown to produce useful representations without using augmentations during training. We pre-train VideoMAE models on our generated datasets and evaluate them on action recognition tasks.

**Learning from synthetic videos.** Synthetic video data is widely used for solving low-level video tasks. Specifically, data generated by 3D simulators (e.g. video game engines), were shown to be useful data sources for training models on optical flow (Dosovitskiy et al., 2015) and point tracking (Zheng et al., 2023) as ground-truth labels for these tasks can be computed from the simulators. Guo et al. 2022 pre-trained a contrastive model on videos generated from a game simulator to learn representations for human motion. Kim et al. 2022 explored how transferable are video representations learned from synthetic video data of public 3D assets. In contrast to these methods, we use only *simple* generative models that aim to mimic *known properties* of natural videos in order to analyze what are the elements that enable useful video representation learning.

**Analyzing dataset curation processes.** The research attention toward curating and characterizing useful pre-training datasets has grown recently. Various approaches were proposed for summarizing the properties of such datasets. Dataset distillation approaches aim to summarize the datasets into a few examples that lead to the same model performance as the original datasets after training (Cazenavette et al., 2022; Wang et al., 2018). Gadre et al. 2024 introduced a benchmark for evaluating different dataset curation processes used for learning downstream tasks. Fang et al. 2024 explored the correlation between data filtering heuristics and downstream performance for image classification, and proposed data filtering networks to improve filtering.

Closer to our work, Baradad et al., 2021; 2022 proposed a progression of generative *image* models for exploring the data properties that can unlock effective model pre-training. We follow a similar approach for video pre-training and propose a progression of generative *video* models. However, unlike Baradad et al., 2021, each model in our progression is built *on top* of the previous model.

## 3 PRE-TRAINING VIDEO MODELS WITHOUT NATURAL VIDEOS

To close the gap between training from scratch and natural video pre-training, and to find the key elements of the data for synthetic video pre-training, we provide a progression of datasets. The dataset are gradually introducing different aspects that appear in video data (e.g. transforming shapes, accelerating shapes). We pre-train SSL models on each of the generated datasets and evaluate them on downstream tasks. In Section 3.1 we present the progression of datasets and describe the generative processes that create them. Then, in Section 3.1, we present the pre-training and downstream evaluation suit.

### 3.1 PROGRESSION OF VIDEO GENERATION PROCESSES

We start by describing the progression of generative models $\{G_i\}$ we use to generate our training datasets. Each model uses a random number generator to sample latent parameters. The latent parameters are used for generating videos - sequences of $T$ frames $f_t \in \mathbb{R}^{H \times W \times 3}, t \in \{1, ..., T\}$. Each consecutive model is built on top of the previous model, by modifying one aspect of it and adding additional calls to the random number generator. Examples of frames sampled from videos in the progression are shown in Figure 1. The models in the progression are described next (see Appendix A.1 for additional hyper-parameters, and the supplementary material for videos).

**Static circles.** Our first video model is of static synthetic images of multiple circles that are copied $T$ times (e.g. $f_t = f_{t+1}$). The frames are generated by positioning multiple overlapping circles on the frame canvas. The color and location of the circles are sampled uniformly at random. Following the Dead Leaves model (Bordenave et al., 2006), the radius is sampled from an exponential distribution, as this distribution resembles the distribution of objects in natural scenes.

**Moving circles.** Starting from randomly positioned circles in the first frame, each assigned a velocity to derive the next frames by modeling the dynamics. Each circle is assigned a random direction and a velocity magnitude that is sampled uniformly from a fixed range. Each circle is assigned a random z-buffer value, according to the order in which it was positioned on the canvas for the first frame. This depth assignment results in occlusions when objects are moving. Introducing changes in the temporal dimension allows us to evaluate the importance of dynamics for video understanding tasks.

**Moving shapes.** We replace the circles sampled for the first frame with different shapes, including circles, quadrilaterals, and triangles. The shape types are sampled uniformly at random, and velocities are applied to them to simulate the next frames, similarly to the previous model.

**Moving and transforming shapes.** We introduce temporal transformations to the sampled shapes and apply them together with the velocities to derive the next frames. Each shape is assigned uniformly at random two scaling factors (one for each spatial dimension), a rotation speed, and two sheer factors. Each consecutive frame is computed by scaling the object in the current frame by the scaling factors, rotating it, and applying the shear mapping.

**Accelerating transforming shapes.** To introduce more complex dynamics, each temporally transforming shape is accelerated during the video by a random factor. The acceleration value is sampled uniformly from a fixed range that includes both positive and negative values.

**Accelerating transforming textures.** We replace the solid-colored shapes from the previous dataset with textures, to integrate realistic image patterns into videos. We utilize synthetic texture images from the statistical image dataset (Baradad et al., 2021). This dataset mimics color distribution, spectral components, and wavelet distribution characteristics of natural images and was shown to be useful for image pre-training. We use a total of 300k textures and for each of the shapes in the previous dataset in the progression, we sample a random texture to replace its solid color.

**Accelerating transforming StyleGAN crops.** We replace the statistical textures with texture crops from the StyleGAN-Oriented dataset (Baradad et al., 2021). This dataset contains 300K texture images that were sampled from an untrained StyleGAN (Karras et al., 2020) initialized to have the same wavelets for all output channels in the convolution layers. It was shown to be the most useful for *image* model pre-training, out of all the synthetic datasets presented in Baradad et al. (2021).

**Accelerating transforming image crops.** We substitute the synthetic textures sampled for the previous Oriented-StyleGAN dataset with natural image crops, taken from ImageNet (Deng et al., 2009). We do not parse or segment the images; instead, we sample random crops in the shapes mentioned above.

## 3.2 PRE-TRAINING PROTOCOL

We study the progression of generative models described above, by pre-training video models on sampled videos from each generator $G_i$ and evaluate them on downstream tasks. This results in a progression of pre-trained models $\{M_i\}$, where $i$ is the index of the dataset in the progression. Next, we describe our choice for pre-training model architecture, dataset sizes, and the baselines we compare to.

**Pre-training model.** We use VideoMAE (Tong et al., 2022) as our pre-training approach. Differently from other masked video auto-encoding approaches presented in Section 2, this method uses tube masking. It has been shown to outperform other SSL methods (e.g. contrastive learning approaches) without relying on heavy augmentations during pre-training. We evaluate the pre-trained encoder of the model by fine-tuning and linear-probing it on downstream tasks. We use different model sizes to verify the consistency of the improvements in performance across scales.

**Baselines.** We compare the pre-trained models to two additional models - a VideoMAE model that was pre-trained with the self-supervised reconstruction objective on the training data of the downstream evaluation data (UCF101), and a VideoMAE model that was initialized with random weights (e.g. trained from scratch). The former can be viewed as an upper bound for our progression, as this model is pre-trained on natural videos from the same distribution as the test set. The latter can be viewed as a lower bound, as no pre-training is done in this baseline.

**Dataset sizes and pre-training hyper-parameters.** We use the same hyperparameters as in the original pre-training recipe. That includes the same number of training steps and fine-tuning/linear probing steps. While we can generate infinite datasets from the generative models we described above, we aim to be comparable to the original pre-training dataset (UCF101). Therefore, for all the generative models that use textures or image crops, we generate sets with a similar size to the original pre-training dataset. For the other datasets, as the model manages to memorize the training data if the size is similar to the pre-training dataset, we generate random examples on the fly.

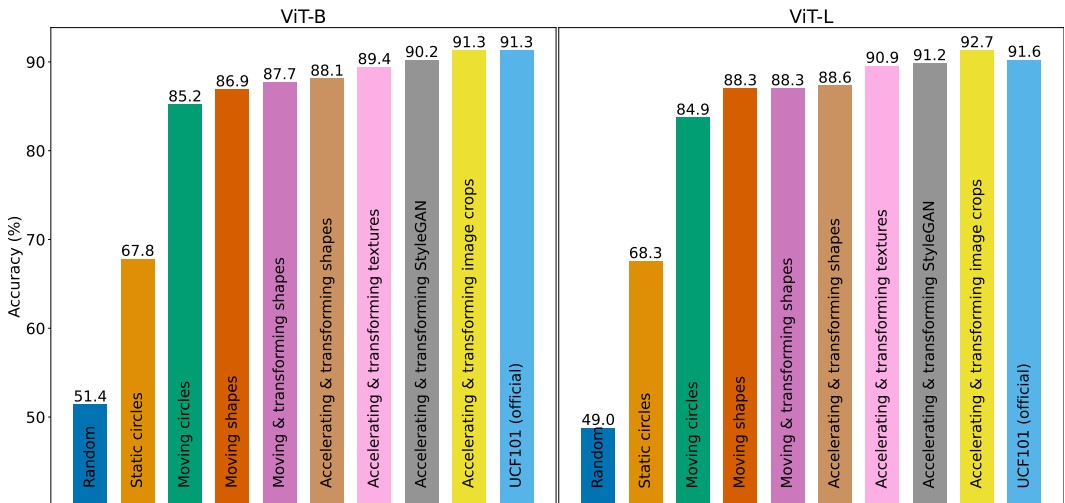

Figure 2: **Action recognition accuracy on UCF101.** We present the UCF101 classification accuracy of the progression of models $\{M_i\}$, after fine-tuning each of them on UCF101. The accuracy increases along the progression.

### 3.3 Evaluation protocols

We evaluate our pre-trained models for action recognition. We test the models on UCF101 (Soomro et al., 2012), a dataset that contains 13,320 video clips of human actions categorized to 101 classes, and on HMDB51 (Kuehne et al., 2011), a dataset of additional 6766 human action video clips categorized to 51 classes. We evaluate out-of-distribution action recognition on UCF101-P (Schiappa et al., 2023), which includes videos from the test-set of UCF101, corrupted with 4 types of low-level synthetic corruptions - camera motion, blur, noise, and digital corruptions.

As mentioned in Section 3.2, each model is pre-trained and fine-tuned for the same number of steps and with the same hyper-parameters (provided in Appendix A.2). The length of the videos and the width and height are sampled to be similar to UCF101. We use the official UCF101 pre-trained checkpoint of VideoMAE as our baseline.

## 4 Experimental Results

We analyze how pre-training on data sampled from the generative models presented in Section 3 affects the downstream performance. We show results for fine-tuned models on in-distribution and out-of-distribution datasets (Sections 4.1 and 4.2) and for linear-probed models (Section 4.3).

### 4.1 Fine-tuning

We fine-tune the pre-trained models for two different model scales, ViT-B and ViT-L, and evaluate the action recognition accuracy on UCF101 and HMDB51. We follow the protocol and hyper-parameters of Tong et al. 2022 and tune only the learning rate and batch size.

**UCF101 action classification.** The results are presented in Figure 2. The final model in the progression, accelerating and transforming shapes with ImageNet crops, performs similarly to the model that was pre-trained on the UCF101 dataset (ViT-B), or outperforms it (ViT-L). Each fine-tuned model $M_i$ in the progression improves over its predecessor, for both model scales. A large increase in performance happens when dynamics are introduced to the generated data (e.g. from static circles to moving circles).

**HMDB51 action classification.** We evaluate the pre-trained models by fine-tuning them on the HMDB51 and present the results for ViT-B in Table 1. As shown, the order of the progression for the classification accuracy is similar. The two last models in our progression are more accurate than the model that was pre-training on UCF101.

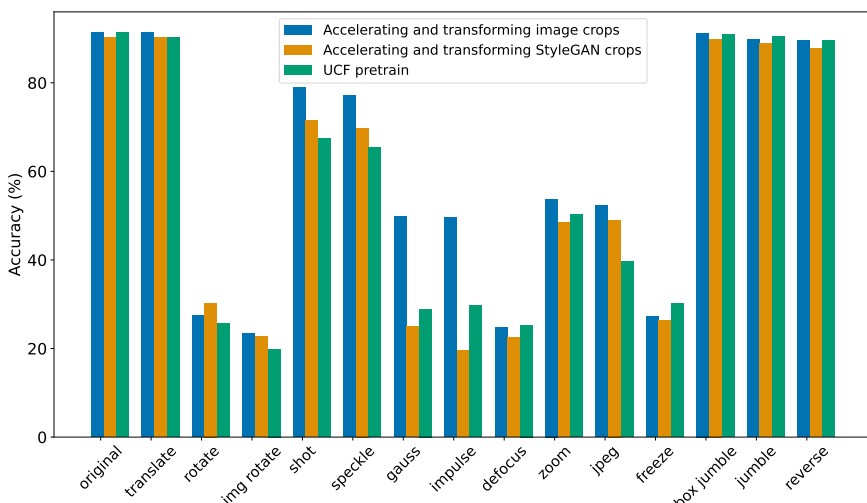

Figure 3: **Distribution Shift results on UCF101-P (Schiappa et al., 2023) (ViT-B)** The last model in our progression outperforms pre-training on natural videos for 11 out of 14 corruption datasets.

**Comparison to synthetic image pre-training for image classification.** The *image* model from Baradad et al. 2021 that achieved the best performance after pre-training on synthetic data and fine-tuning on ImageNet classification task (Deng et al., 2009), has an accuracy of 74.0%. Compared to the baseline model that was randomly initialized (with an accuracy of 60.5% after fine-tuning), and to a model that was pre-trained on real ImageNet data (with an accuracy of 76.1%), the model trained on the synthetic data closes 86.5% of the gap. For UCF101, our ViT-B model closes 97.2% of the gap when using crops from the StyleGAN synthetic dataset, and reaches the same accuracy as the UCF101 pre-trained model with image crops (with the same size datasets as the pre-training data). This suggests that, unlike image SSL models, current video SSL models do not utilize the natural data efficiently and that most of the performance can be recovered by training on synthetic data coming from simple generative processes.

## 4.2    DISTRIBUTION SHIFT

We fine-tune the pre-trained models $\{M_i\}$ on UCF-101, and evaluate on corrupted datasets from UCF101-P (Schiappa et al., 2023). The results for the last two models in the progression are presented in Figure 3. As shown, the last model in the progression outperforms the UCF101 pre-trained model on 11 out of 14 tasks, and performs comparably on the rest. This suggests that while the current pre-train recipe fails to generalize to out-of-distribution datasets. We note that the second to last model in our progression, that does not use real images, performs better only on 6 out of the 14 datasets. This suggests that differently from StyleGAN textures, the natural image crops unlock generalization capabilities to out-of-distribution *video* corruptions.

## 4.3    LINEAR-PROBING

We linear-probe the progression of pre-trained models on UCF101. We use the same hyper-parameters as used for fine-tuning, replacing only the base learning rate to $0.01$, and do not use weight decay. The results are presented in Table 1.

**Comparison between fine-tuning and linear-probing results.** There are two main differences in the results after linear probing when compared to the results after fine-tuning. First, the difference in performance between the last model in the progression and the model trained on UCF101 is more significant (a gap of 23.2%). Compared to the best model of Baradad et al. 2021 that was trained on synthetic image data, which closes 56.5% of the gap between linear probing on randomly initialized weights and linear probing on a pre-trained model, the last model in our progression closes only

| | HMDB51 fine-tune | UCF101 lin. prob | UCF101 fine-tune |
|---|---|---|---|
| Random initialization | 18.2 | 8.9 | 51.4 |
| Static circles | 29.2 | 13.2 | 67.8 |
| Moving circles | 52.0 | 15.5 | 85.2 |
| Moving shapes | 56.1 | 20.4 | 86.9 |
| Moving and transforming shapes | 57.6 | 18.8 | 87.7 |
| Acc. and transforming shapes | 58.9 | 18.9 | 88.1 |
| Acc. and transforming textures | 62.4 | 20.9 | 89.4 |
| Acc. and transforming StyleGAN crops | **64.1** | 25.2 | 90.2 |
| Acc. and transforming image crops | **64.1** | 24.8 | **91.3** |
| UCF101 | 63.0 | **48.0** | **91.3** |

Table 1: **Additional action recognition results (ViT-B).** We present the classification accuracy on HMDB51 after fine-tuning and on UCF101 after linear probing/fine-tuning for all the pre-training datasets in our progression and the two baselines.

40.6% of the gap. We suspect that the difference in the gap between fine-tuning and linear probing is due to large differences between low-level properties of natural images and our datasets, which can be mitigated by fine-tuning the full model. We analyze these low-level properties in Section 5.4.

The second difference is that there is the progression order (when sorting by accuracy). Specifically, in contrast to fine-tuning (both on HMDB51 and UCF101), introducing gradual transformations to the shapes decreases the linear probing performance of the model compared to the previous dataset. Moreover, the order between the rest of the consecutive models in the progression is different, although the differences in performance are small. Finally, the model that uses synthetic StyleGAN crops performs better than the last model in the progression.

# 5 DATASETS ANALYSIS

In this section, we analyze in depth a few characteristics of the synthetic datasets that were shown to be useful for video pre-training. We start by evaluating the effect of incorporating natural images in the training. Then, we analyze the effects of different types of synthetic textures. Finally, we compare the statistical properties of videos to the downstream performance.

## 5.1 INCORPORATING STATIC IMAGES

Following the improvement of the model performance when natural image crops are used in the pre-training data, we raise three questions: 1) how does the size of the static image dataset affect the downstream performance, 2) can the pre-training benefit from both synthetic and natural texture crops, and 3) are there alternative ways to incorporate natural images in the pre-training regime? Next, we address these questions.

**Image dataset size.** We evaluate the effect of the image data size on the downstream task. Our initial pool of images includes all the images from ImageNet (1.3M). We provide additional results with a pool with 300k images, while keeping the size of the pre-training video dataset fixed. We use the same acceleration, speed, and shape transformations as in the last dataset in the progression. The results for ViT-B, fine-tuned for the UCF101 classification task, are presented in Table 2. An increase in the size of the static images dataset results in a better performance on the downstream task.

**Combining natural images and synthetic textures.** To evaluate if useful pre-training can be achieved by combining natural images and synthetic textures, we create a dataset that incorporates crops from half of the images and crops from half of the synthetic textures from the StyleGAN textures (Baradad et al., 2021) that we used in the previous dataset in the progression (each has 150k examples). We apply the same acceleration, speed, and transformations as in the last dataset in the progression. As shown in Table 2, the performance of the new dataset ("150k images & 150k StyleGAN") is slightly higher than the performance of the two datasets that use solely one type of data. This suggests that mixing datasets can lead to improved performance in other cases as well. We leave this approach to future work.

| Configuration | Accuracy (%) |
|---|---|
| 300k images | 90.5 |
| 150k images & 150k StyleGAN | 90.6 |
| 300k StyleGAN | 90.2 |
| 300k statistical textures | 89.4 |
| 1.3M images | 91.3 |
| Replacing 5% of videos w/ static images | 88.5 |

| Configuration | Accuracy (%) |
|---|---|
| Static StyleGAN crops | 90.2 |
| Dynamic StyleGAN crops | 89.2 |
| Dynamic StyleGAN videos | 68.7 |

Table 2: **Incorporating natural images into training (ViT-B).** We ablate different approaches for incorporating natural images during training, and evaluate them on UCF101.

Table 3: **Incorporating synthetic textures into training (ViT-B).** Introducing dynamics to the StyleGAN textures does not improve performance.

**Mixing static videos of repeating single images.** We present an alternative approach to incorporate natural images into the dataset - instead of cropping images, we use full images and create videos from them by repeating the same image across all the frames. We append these static videos to the "Accelerating and transforming shapes" dataset, to make them the only source of textures. Their ratio in the mixed dataset is 5% (as we found this ratio to be optimal for downstream tasks).

While the downstream model performs better than the model that was trained on the "Accelerating and transforming shapes" dataset (see "Replacing 5% of videos w/ static images" in Table 2), the model performs worse than using texture crops or image crops.

## 5.2 INCORPORATING TEXTURES

While the best model in our progression uses image crops, we seek other alternatives with synthetic textures. Specifically, we replace the static StyleGAN textures with a dynamic version.

**Dynamic StyleGAN textures.** We investigate a simple extension of the StyleGAN-generated textures into videos. We create a texture video by starting from a random noise $z_0$, provided as a latent code to the StyleGAN generator $G'$. Each consecutive frame is generated by adding a random noise with a smaller standard deviation $\delta z_i$ to the previous latent $z_{i-1}$ ($z_i = z_{i-1} + \delta z_i$) and generating a frame $G'(z_i)$. We explore two approaches to incorporate these texture videos: directly creating a dataset with multiple such videos ("Dynamic StyleGAN videos") or replacing the solid-color shapes from the "accelerating and transforming shapes" with dynamic texture crops that are updated across frames ("Dynamic StyleGAN crops").

**Fine-tuning on UCF101.** Table 3 presents the action classification accuracy after incorporating the dynamic textures in the pre-training stage and fine-tuning on UCF101. Using videos of random walks in the latent space of randomly initialized StyleGAN leads to performance that is only slightly better than training on static circles (67.8%). Replacing the static StyleGAN crops with *dynamic* StyleGAN crops leads to a performance drop of 1%. That suggests that the simple hand-crafted dynamics of randomly moving Dead-Leaves models are sufficient for pre-training, without the need for introducing additional dynamics modeling.

## 5.3 SIMILARITY TO PRE-TRAINING DATASET

During our experiments, we created multiple versions of each dataset we presented, with differences in configuration (e.g. different video background colors and different object speeds). In total, we generated 28 datasets and trained ViT-B VideoMAE on each. We plot the UCF101 fine-tuning accuracies of the models as a function of their similarity to UCF101. We present two similarity metrics - video similarity and single frame similarity (FID Heusel et al. (2017)).

**FID.** We compare the similarity between *frames* from our datasets and UCF101 to the classification accuracy. We compute FID (Heusel et al., 2017) on randomly sampled frames (10k frames from each dataset). FID is a common metric for evaluating the similarity between two image datasets by comparing the Frechet distance between distributions of deep features extracted from them.

As shown in Figure 4.a there is a strong negative correlation between the frame similarity to the accuracy ($r = -0.72$). This suggests that improving frame similarity can lead to better performance.

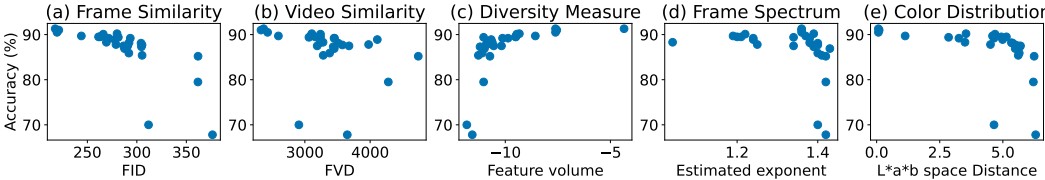

Figure 4: **Dataset properties compared to downstream performance.** We compare the downstream classification accuracy on UCF101 after fine-tuning to frame and video properties of all the dataset variants we used in our analysis (see datasets list in Appendix A.1).

Nevertheless, the FID scores are considered to be high, and our datasets are significantly different from the original UCF101 data.

**FVD.** In this analysis we compare the classification accuracy to the *video* similarity between our datasets and UCF101. We compute FVD (Unterthiner et al., 2019) on 1,000 random videos from each of the datasets and present the results in Figure 4.b. Differently from the frame similarity, there is less significant negative correlation between the FVD metric and the performance ($r = -0.27$). This suggests that this metric is less indicative of downstream performance.

## 5.4 STATIC PROPERTIES OF INDIVIDUAL FRAMES

We follow Baradad et al. 2021 and compare the properties of individual frames in the 28 datasets that we generated to the downstream performance. Similarly to Section 5.3, we randomly sample 1000 videos from all the datasets we analyzed and compare low-level statistics to the downstream classification accuracy.

**Diversity.** We follow Baradad et al. (2021), and measure the diversity of the frames in the dataset. We utilize inception features (Szegedy et al., 2015) computed for 16 sampled frames in randomly sampled videos and plot the determinant of their covariance matrix. The results are presented in Figure 4.c. There is a moderate correlation between the accuracy and the diversity ($r = 0.53$). According to this measure, all the generated datasets are less diverse than UCF101. Nevertheless, the datasets that include synthetic textures (statistical or StyleGAN-based) and the ones that include image crops are more diverse than the other datasets. This suggests that investing in more diverse datasets can improve performance even further.

**Image spectrum.** Following Torralba & Oliva 2003, that showed that the spectrum of natural images resembles the function $A/|f|^\alpha$, with a scaling factor $A$ and an exponent $\alpha$ ranging in $[0.5, 2.0]$, we estimate the exponent for frames in our datasets. The results are presented in Figure 4.d. The datasets that result in the best downstream performance have an estimated exponent that lies close to the middle of the range, between 1.2 and 1.4.

**Color statistics.** We compare the distance in color space between the generated data and natural videos. Similarly to Baradad et al. 2021, we compute the symmetric KL divergence between the color distributions of each dataset. We model the color distributions as three-dimensional Gaussian that correspond to the three color channels in L*a*b space. Figure 4.e presents the distances between UCF101 color statistics and the datasets in our progression. There is a relatively weak negative correlation of $r = -0.42$ between the color distance to UCF101 and the accuracy.

## 5.5 REPRESENTATION VISUALIZATION

We visualize the learned representation produced by the models $M_i$. Following Amir et al. 2023, we compute PCA on the attention keys extracted from the last VideoMAE encoder layer across 32 frames from 70 videos from the same class of UCF101. We plot the first three principal components as red, green, and blue channels and present features for 2-frame inputs (see temporal PCA for full videos in the supplementary material).

The visualizations for videos from three classes are presented in Figure 5. The principal components of the features produced by the pre-trained models are relatively different. While the early models in the progression capture mostly static positional information about the frames, later models preserve some structural information in the input in the 3 principal components.

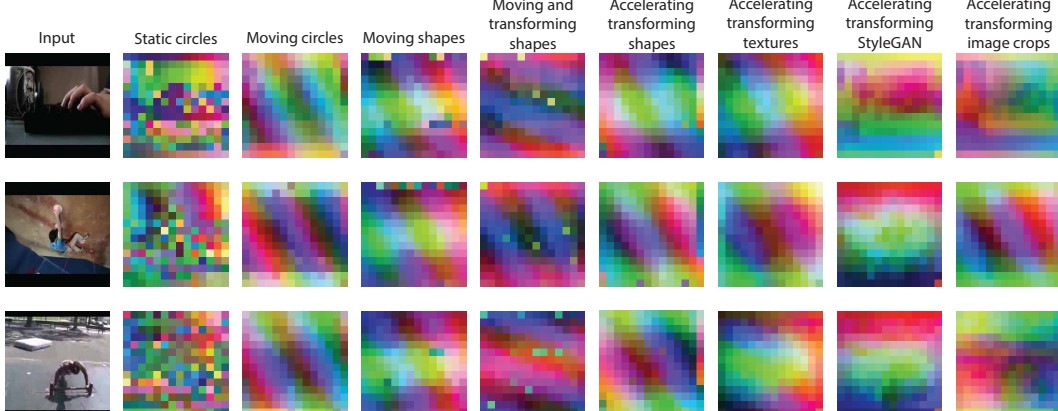

Figure 5: **Feature visualizations for pre-trained models.** We present the 3 principal components of the attention keys of the last encoder layer, for all $M_i$ as the three color channels. Different object parts start to appear as the datasets progress.

## 6 LIMITATIONS AND DISCUSSION

We conclude by presenting three limitations of our analysis and discussing future work.

**Generalization to other tasks.** While the pre-trained models are evaluated on three different datasets (HMDB51, UCF101, and UCF101-P) and with two different adaptation regimes (linear probing and fine-tuning), and show relatively similar trends across the progression of the datasets, they can have different trends when adapted to other tasks. We decided to focus on action recognition and to aim to reach the performance of relatively *small datasets*, as a first step to a fully synthetic approach that does not rely on natural videos. We do not tune any hyper-parameters (except batch size and learning rate, due to GPU memory capacity differences) to improve performance. In future work, we aim to extend this approach to other tasks and apply training regimes that were shown to work on larger datasets, hoping to reach similar performance.

**Generalization to other model types.** Our evaluating suite included pre-training of one type of model - VideoMAE. While this pre-training approach is widely used, the behavior we presented for different datasets may be different for other pre-training regimes. Our decision to focus on one model follows a similar scope of Baradad et al. 2021.

**Properties of the mixed image datasets.** We show that static image data that can be used as crops during the pre-training stage can improve downstream performance. While we show that *more images* result in better performance, our analysis does not answer *what type* of natural image data is useful for video pre-training. We plan to explore this question in future work.

**Discussion.** Learning from data produced by simple generative processes and other well-studied data sources has an advantage over learning from large-scale video data - when pre-training on large-scale video corpus, commonly obtained from the internet, it is merely impossible to monitor what are all the training examples and to verify that no malicious, private, or biased data is included in the pre-training stage. Learning from generated data, on the other hand, gives better control over the type of data that is provided during pre-training.

We believe that the synthetic data analysis we provided can be utilized to create better datasets for learning video representations without natural videos. Guided by this analysis, we plan to investigate other well-understood data sources and generation processes to continue improving video representation learning, in large-scale training regimes.

While it was not our aim in this paper, the synthetic data we produced can be incorporated as augmentations as well. Pre-training on UCF101 *together with* the last data in the progression leads to accuracy of 92.% after fine-tuning ViT-B VideoMAE, surpassing the performance of UCF101 pre-training. We plan to explore this direction in the future as well.

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

# A  APPENDIX

## A.1  ADDITIONAL DATASET DETAILS

We provide the hyper-parameter configuration for the dataset generators in Table 7. We provide additional explanations next. Please see the attached HTML for videos of the datasets in our progression.

**Dataset size.** For datasets without textures or image crops, we use an on-the-fly generation strategy for training. For video data with textures and image crops, we generate 9537 videos for training, the same number in the UCF101 training set. For all each generated video, we use a resolution of $256 \times 256$ for, FPS of 25, and a duration that is sampled uniformly in $(100, 200)$.

**Acceleration and speed parameters.** For each object, we sample its absolute speed from a uniform distribution ranging between $1.2$ and $3, 0$ pixels per time frame. The absolute acceleration sampled from a uniform distribution $(-0.6, 0.6)$. The moving direction sampled uniformly between $(-\pi, \pi)$.

**Transformation parameters.** To introduce dynamics additionally to translation, we apply scale, shear, and rotation transformations. By default, the rotation angle is set to a uniform distribution of $(-\frac{1}{100}\pi, \frac{1}{100}\pi)$, scale and shear factors are set to a randomly chosen number from $(-0.005, 0.005)$ in both x-axis and y-axis.

## A.2  TRAINING CONFIGURATION

We provide the hyperparameter configurations for pre-training (Table 4), fine-tuning on UCF101 (Table 5), and linear probing (Table 6) on the ViT-B model. The configuration for fine-tuning is similar to the original configuration from Tong et al. 2022, except for the batch-size, learning rate, and the Adam optimizer hyper-parameters. The fine-tuning configuration on HMDB51 is the same as for UCF101, except for the number of test clips, which is 10. The pre-training and fine-tuning setting for ViT-L is same with ViT-B, except for reducing batch size to half.

| Hyperparameter | Value |
|---|---|
| masking ratio | 0.75 |
| training epochs | 3200 |
| optimizer | AdamW |
| base learning | 3e-4 |
| weight decay | 0.05 |
| optimizer momentum | $\beta_1 = 0.9, \beta_2 = 0.95$ |
| batch size | 256 |
| learning rate schedule | cosine decay |
| warmup epochs | 40 |
| augmentation | MultiScaleCrop |

Table 4: **Pre-training settings (ViT-B).**

## A.3  ADDITIONAL GENERATED DATASETS

Apart from the progressions presented in the main paper, we explored video dataset properties from other perspectives. including but not limited to object dynamics, textures information, frame diversity and real data usage. We provide a brief description of additional datasets below.

- **Moving objects with slower speed**: For this family of datasets, We repeat some of the progressions mentioned in the main paper, but with slower movement (50% of the speed in main progression) and study how the velocity affects the temporal information. The datasets used for this setting includes moving circle, moving shape, moving and transforming shape, accelerating transforming shape and accelerating transforming textures. We present the results of datasets with slower dynamics in Table 8.

| Hyperparameter | Value |
|---|---|
| training epochs | 100 |
| optimizer | AdamW |
| base learning | 1e-3 |
| weight decay | 0.05 |
| optimizer momentum | $\beta_1 = 0.9, \beta_2 = 0.95$ |
| batch size | 256 |
| learning rate schedule | cosine decay |
| warmup epochs | 5 |
| flip augmentation | yes |
| RandAug | (9, 0.5) |
| label smoothing | 0.1 |
| mixup | 0.8 |
| cutmix | 1.0 |
| drop path | 0.2 |
| dropout | 0.0 |
| layer-wise lr decay | 0.7 |
| test clips | 5 |
| test crops | 3 |

Table 5: **Fine-tuning settings (ViT-B)**

| Hyperparameter | Value |
|---|---|
| training epochs | 100 |
| optimizer | AdamW |
| base learning | 1e-2 |
| weight decay | 0.0 |

Table 6: **Linear probing settings (ViT-B)**

- **More texture types**: As discussed in Section 5.2, we studied different textures settings. In addition to the results in Table 3 and Table 2, we generated some other textures related data for better understanding. The results are shown in Table 9

  - **Dynamic StyleGAN high-freq**: A less diverse StyleGAN generator from Baradad et al. (2021), with only high grequency noise as input to build image structure. We make a new dataset from it by gradually adding random noise as mentioned in main paper.

  - **Replacing with statistic videos from StyleGAN**: Same as the last setting in Section 5.1, we also replace 5% of the accelerating transforming shapes into StyleGAN samples, which are repeated 16 times to mimic a video.

  - **150k images and 150k statistical textures**: we create a dataset that incorporates crops from half of the images and crops from half of the statistical textures we used in the previous dataset in the progression. We apply the same operation in this dataset as in main progression.

- **More diverse background**: To introduce more diversity into the video, we try to replace the default black background with more diverse and semantic meaningful images. The results are shown in Table 9

  - **Image crops, with colored background**: We took the same generation setting from '300k images' in the Table 2. For each video, instead of black video, we random sample a color and use as background.

  - **Image crops, with image background**: Same as the setting above, except that we use a random image from the image crops set to serve as background in each video.

| Hyperparameter | Value |
|---|---|
| Initial speed range | (1.2, 3.0) |
| Acceleration speed range | (-0.06, 0.06) |
| Rotation speed range | $(-\frac{1}{100}\pi, \frac{1}{100}\pi)$ |
| Scale X speed range | (-0.005,0.005) |
| Scale Y speed range | (-0.005,0.005) |
| Shear X speed range | (-0.005,0.005) |
| Shear Y speed range | (-0.005,0.005) |

Table 7: **Dataset generation settings**

- **Real data mixture**: Given the powerful ability of synthetic data, we aim to find out if real data and synthetic data can boost each other in downstream task. The accuracy on UCF101 fine-tune setting is presented in Table 10.

  - **Accelerating and transforming textures, mix with real video data**: We try replacing 25% and 75% of training data by sampling real videos from UCF101 training set.

  - **50% imagenet crops and 50% UCF101**: We create a new data set by randomly sampling from last progression in main paper, and the UCF101 dataset. We make sure that the sample rate is 1:1 and training size is same as standard experiments.

- **Saturated textures**: During exploration, we create a different set of textures-based datasets by making a saturated color version of the datasets. For each moving object, we sampled a random color and added on the texture crops. Surprisingly, despite the the possible corruption in the texture information, they still presents competitive performance. A full list of color saturated datasets and present the results in Table 11

| Dataset configuration | UCF101 |
|---|---|
| Moving circles | 84.9 |
| Moving shapes | 88.3 |
| Moving and transforming shapes | 88.3 |
| Accelerating and transforming shapes | 88.6 |
| Accelerating and transforming textures | 90.9 |

Table 8: **Additional datasets (ViT-B).** Moving objects with slower speed

| Dataset configuration | UCF101 |
|---|---|
| Dynamic StylaGAN high-greq | 68.7 |
| Replacing 5% of videos w/ StyleGAN | 88.2 |
| 150k images & 150k statistical textures | 89.7 |
| 300k images w/ colored background | 89.9 |
| 300k images w/ image background | 91.0 |

Table 9: **Additional datasets (ViT-B).** More texture types and more diverse background

| Dataset configuration | UCF101 |
|---|---|
| Accelerating and transforming shapes, 25% w/ UCF101 | 90.4 |
| Accelerating and transforming shapes, 75% w/ UCF101 | 90.6 |
| Accelerating and transforming image crops, 50% w/ UCF101 | 92.0 |

Table 10: **Additional datasets (ViT-B)**. Mix with real videos

| Dataset configuration | UCF101 |
|---|---|
| Statistical textures | 88.9 |
| Statistical textures w/ colored background | 87.8 |
| Moving Dynamic StyleGAN crops | 87.5 |
| 300k image crops | 90.1 |
| 150k image crops & 150 statistical textures | 89.2 |
| 300k image crops w/ colored background | 89.5 |
| 300k image crops w/ image background | 89.5 |
| 1.3M image crops | 89.8 |

Table 11: **Additional datasets (ViT-B).** Saturated textures

