# OpenReview forum: "Video Representation Learning Without Natural Videos"
_ICLR.cc/2025/Conference — ICLR 2025 Conference Withdrawn Submission_

### Official Review · Reviewer_ZAze · 2024-10-29

**Soundness:** 1
**Presentation:** 2
**Contribution:** 2
**Rating:** 1
**Confidence:** 4

**Summary:**

The authors construct a synthetic video dataset focused on various motions and describe the dataset construction process in detail. Then they train a VideoMAE model on this synthetic dataset and compare against a baseline trained on a smaller natural video dataset. They also evaluate two video classification datasets and on an out-of-distribution (augmented) variant of one dataset. Authors claim competitive performance of their method and next analyze which aspects of the synthetic datasets contributed to the strong performance.

In the dataset analysis, they examine benefits of incorporating natural images and synthetic textures. Larger image datasets and a mix of natural and synthetic textures improve downstream performance. Static synthetic textures outperform dynamic ones, and datasets with higher frame similarity and diversity yield better results. Color and spectrum properties also moderately impact performance. They also visualize PCA of model attention maps to show the model’s ability to capture structural information.

**Strengths:**

1) The idea of using purely synthetic data for video SSL in interesting, and if thoroughly investigated could provide valuable insights regarding video SSL techniques.
2) The synthetic dataset generation process is well described and thorough with clear motivation for each of its variants. This can be useful for other works exploring similar directions.
3) Statistical analysis of the generated dataset could be useful but is not presented clearly.

**Weaknesses:**

**1) Lacking understanding of prior work in video self-supervised learning:**

a) “Specifically, in the video domain, although various large-scale datasets exist and have been incorporated via similar self-supervised learning tasks, the improvements in downstream performance on video understanding (e.g. action recognition) are relatively low.” - this is not true, video SSL has lead to clear improvements on action recognition over non-video SSL or supervised pre-training initializations [1, 2, 3]. These improvements are on par with image SSL methods as established in prior work [1, 2, 3].

b) A large body of contrastive SSL for video is ignored in related work. See related works sections in [1, 2, 3] for relevant papers.

**2) Possibly incorrect problem formulation:**
In L204-209, the videoMAE model pre-trained on UCF-101 is viewed as an upper bound (since this is the test data distribution). However, SSL techniques behave differently (i.e. poorly) on smaller datasets as opposed to larger ones, even if the small dataset is the test distribution. See Table 2 in [4] where UCF-101 results are 4-6% points higher when pre-trained on Kinetics-400 dataset instead of UCF-101 itself. This is further validated by later works exploring video SSL focussed on Kinetics-400 to avert such issues [1, 2, 3]. The same behavior can be seen in VideoMAE paper [5] when comparing Table 2 vs Table 3.
The authors use this assumption of an upper bound for all comparisons, claiming to “close 97.2% of the gap to this upper bound” (see L052 in intro). However, this upper bound is much lower than the actual best performance of these SSL methods.

**3) Unfair experimental setup:**
The upper-bound baseline is trained only on UCF-101 (which as mentioned in point 2 is already a handicapped version of these SSL methods). Adding to this, UCF-101 has only 8.5K images while their synthetic dataset used to train models has 300K-1.3M images (35 to 140 times more data). This makes the numbers reported in Table 1 clearly unfair. In such a case where different datasets are used, the authors should at least report the sizes of the dataset. The current style of reporting these results appears almost intentionally misleading the reader. For a good example, see Table 2 in [5] which the authors themselves use as their baseline where given different pre-training datasets, their sizes are reported.

**4) Linear Probing Baselines:**
It is known that linear probing with VideoMAE results in subpar performance. Contrastive methods such as DINO are better for linear probing. See results in [1, 6] where UCF-101 linear probing achieves over 90% accuracy. The 24.8% accuracy with the synthetic dataset reported in Table 1 does not by any means support the authors claim of “that useful video representations can be learned from synthetic videos and natural images, without incorporating natural videos in the training.”

**5) Evaluation Task Mismatch:**
The synthetic dataset construction specifically focuses on injecting various motions into the data. However, solving the downstream tasks (UCF / HMDB) do not require much motion awareness. See [6, 7] where single frame classifiers achieve 85% / 49% accuracy on these datasets. In [6], zero-shot CLIP [8] (with no video training) achieves 69% / 46% accuracy with only a single frame on these datasets.
Consider exploring motion focussed datasets such as SSv2, Diving48, FineGym. The synthetic dataset may actually contribute to mote noticeable improvements in such cases and it is known how mist video SSL methods are relatively weaker at such motion heavy datasets (especially under linear probing settings) [2, 5].

**6) Insufficient baselines:**

a) SSL techniques: The authors cite [9] to support their decision to use only videoMAE as a baseline. However, at the time of writing that paper contrastive methods were clearly dominant in video SSL over other approaches. In contrast, currently these are several contrastive, MAE style, and predictive style SSL techniques all performing competitively on video benchmarks. Therefore it is apt to evaluate this method on at least on more baseline, especially given how the authors chose to evaluate under linear probing settings where MAE approaches are known to perform poorly sometimes.

b) SSL dataset: Even in the referred work [9], their contrastive baseline is trained on multiple datasets (that are large scale and different to the smaller test datasets). It is the same case in the original VideoMAE paper (SSL on several datasets). The authors should train their SSL baseline on at least one large dataset following prior work.

**7) Statistical Analysis of Dataset (minor):**
The procedure and efforts are good, but since the accuracy numbers are unreliable, it is unclear if these findings actually make sense (i.e. are they actually helping video SSL). Figure 4 could provide more information clearly: maybe color the variants in different colors?

**8) PCA visualization (minor):**
It is unclear what the colors are; is this the distance to each of the top 3 principal components? What is dark vs light?
Alternately, consider a visualization such as in the DINO paper (where you group using PCA) and show the 3 different masks (maybe as an image overlay). This could clearly show how attention maps align with image structure.



**References**

[1] Recasens, Adria, et al. "Broaden your views for self-supervised video learning." Proceedings of the IEEE/CVF international conference on computer vision. 2021.

[2] Ranasinghe, Kanchana, et al. "Self-supervised video transformer." Proceedings of the IEEE/CVF Conference on Computer Vision and Pattern Recognition. 2022.

[3] Hu, Kai, et al. "Contrast and order representations for video self-supervised learning." Proceedings of the IEEE/CVF International Conference on Computer Vision. 2021.

[4] Han, Tengda, Weidi Xie, and Andrew Zisserman. "Self-supervised co-training for video representation learning." Advances in neural information processing systems 33 (2020): 5679-5690.

[5] Tong, Zhan, et al. "Videomae: Masked autoencoders are data-efficient learners for self-supervised video pre-training." Advances in neural information processing systems 35 (2022): 10078-10093.

[6] Ranasinghe, Kanchana, et al. “Language-based Action Concept Spaces Improve Video Self-Supervised Learning.” Advances in Neural Information Processing Systems. 2023.

[7] Li, Junnan, Silvio Savarese, and Steven CH Hoi. "Masked unsupervised self-training for label-free image classification.” ICLR 2023

[8] Radford, Alec, et al. "Learning transferable visual models from natural language supervision." International conference on machine learning. PMLR, 2021.

[9] Baradad Jurjo, Manel, et al. "Learning to see by looking at noise." Advances in Neural Information Processing Systems 34 (2021): 2556-2569.

**Questions:**

* Why did the authors use a dataset 35-140 times smaller to train the baseline?
* Why did the authors not follow the K400 baseline numbers in the original VideoMAE paper as the upper-bound?
* Why is there no discussion on recent contrastive video SSL works?

---

### Official Review · Reviewer_PqHH · 2024-11-02

**Soundness:** 3
**Presentation:** 3
**Contribution:** 2
**Rating:** 5
**Confidence:** 4

**Summary:**

The paper presents a method for pretraining video models using a progression of synthetic datasets, gradually incorporating video properties like motion, shape transformations, and acceleration. Starting with static shapes, the authors progressively introduce complexity in the synthetic datasets, culminating in textures and crops from StyleGAN and natural images. This synthetic-only pretraining approach for VideoMAE fills 97.2% of the performance gap on UCF101 compared to models pretrained on natural videos, and outperforms on HMDB51. The authors also examine performance on UCF101-P for robustness and analyze dataset properties to identify correlations with downstream task success.

**Strengths:**

- The approach is novel as it leverages fully synthetic video data for self-supervised learning of video representations, an area that has not been widely explored for video models.

- Pretraining with synthetic datasets achieves comparable performance to natural video pretraining, closing nearly 97% of the performance gap on UCF101 and even outperforming natural video pretraining on HMDB51. The model demonstrates strong robustness, as it outperforms UCF101-pretrained models on 11 out of 14 datasets in the UCF101-P suite, showing the potential of synthetic pretraining for generalization across challenging datasets.

- The paper includes a detailed analysis of the synthetic dataset’s properties, especially the types of textures and natural image crops that are most beneficial, providing valuable insights for optimizing synthetic datasets for video model pretraining.

- The paper is well-structured, clearly written, and easy to follow, with informative figures and logical progression through the methodology, making it accessible for readers.

**Weaknesses:**

- Selection of datasets: this work is mostly focused on two datasets, ucf101 and hmdb51. these datasets are known to have appearance bias, and even image based models have shown very high performance. Therefore conclusions based on just these two datasets are not convincing for learning video representations. The authors should focus on widely used datasets with temporal aspects, such as something-something, diving, etc. [R1]

- Also, the used datasets are small in size and most recent works in video action recognition are focused on large-scale datasets such as Kinetics variants [R2].

- Selection of approach: The authors mainly focus on VideoMAE, and conclusions based on just one approach can not be generalized. Video SSL is an active area of research and the authors should consider some other recent approaches for video SSL [R3, R4].

- Motivation: Another aspect which should be discussed is; why we need synthetic dataset for pre-training? We already have unlimited natural videos available which can be potentially used; what are the advantages of using synthetic videos? It will be good to cover this aspect, otherwise the motivation of this work is weak.

**Questions:**

The main question this work attempts to answer is  L041:"we ask if natural videos are even needed to learn video representations that are similar in performance to current state-of-the-art representations". However, this work fails to answer this question due to the use of only one ssl approach and UCF/HMDB datasets. The experiments performed in this work are not sufficient to be able to answer this question.

The claim L043:" In this work, we reach a downstream performance that is similar to the performance of models
pre-trained on natural videos, while pre-training solely on simple synthetic videos and static images", may be correct to some extent for UCF/HMDB/MAE space, but definitely not enough to make any conclusions, since it was explored for a single technique and datasets with small size and where temporal aspects are not really important.

Some other questions/concerns I have:

- Table 1: poor linear probe performance is not good, as linear probe is preferable over fine-tuning due to its efficiency and preserving learned weights.

- Some more efforts in dataset progression will further strengthen this work.
        - Was accelerating shapes also experimented with? What about number of shapes/circles? Density/size of shapes?
	- Why was circle chosen as a first step? What about square, rectangle, triangle, ete.?
	- Are there complex shapes too?
	- What about static complex shapes?

- Its stated that the size of synthetic dataset is kept same as original datasets, but from Table 2 it seems a large number of natural images have been used (9K ucf training videos vs 1.3M static images), which is a concern since these many images will provide a lot of variations in texture and patches in comparison with original training videos.

- Results on UCF101-DS are not shown; it is a real-world distribution shift dataset. Also, how about results on HMDB51-P, UCF101-P, Kinetics400-P, and SSv2-P?

- Table 1: why linear probe results are missing for hmdb?

- There is no analysis on the impact of number of pre-training videos on the down-stream performance. It will be good to see how the performance varies with the size of pre-training dataset size.

- In Section 5.3, please check if it should it be ‘FID’ instead of ‘frame similarity’ in line 431? : “there is a strong negative correlation between the FID and the accuracy”.

References:

[R1] Goyal, Raghav, et al. "The" something something" video database for learning and evaluating visual common sense." Proceedings of the IEEE international conference on computer vision. 2017.

[R2] Kay, W., Carreira, J., Simonyan, K., Zhang, B., Hillier, C., Vijayanarasimhan, S., ... & Zisserman, A. (2017). The kinetics human action video dataset. arXiv preprint arXiv:1705.06950.

[R3] Schiappa et. al. "Self-supervised learning for videos: A survey." ACM Computing Surveys 55.13s (2023): 1-37.

[R2] Kumar et al. "A Large-Scale Analysis on Self-Supervised Video Representation Learning." arXiv e-prints (2023): arXiv-2306.

---

### Official Review · Reviewer_dyKo · 2024-11-04

**Soundness:** 3
**Presentation:** 3
**Contribution:** 2
**Rating:** 5
**Confidence:** 4

**Summary:**

The paper proposes a method for effectively learning video representations from synthetic videos without the need for training on natural videos. It introduces simple generation steps, such as moving, transforming, and accelerating shapes, to create a series of synthetic video datasets. The authors explore the performance of models pre-trained on these synthetic datasets in downstream tasks, focusing primarily on action recognition using VideoMAE. Experiments on the UCF101 and HMDB51 datasets show that the performance of models pre-trained on synthetic data is comparable to those pre-trained on natural data. Additionally, results from UCF101-P demonstrate that models pre-trained on synthetic data exhibit similar robustness.

**Strengths:**

The proposed method generates synthetic data through operations on shapes (e.g., accelerating, transforming), leading to models that perform comparably to those trained on natural data in downstream tasks. This approach is relatively simple and novel, distinguishing itself from previous work centered on human-based synthetic data.

**Weaknesses:**

1. The paper lacks completeness:
   - a) It primarily focuses on synthetic data for action recognition (Page 6, Line 511). Still, it does not discuss or compare its experiments with similar works by [2] and [3]. Furthermore, the models and datasets used (VideoMAE and UCF101/HMDB51) are less robust compared to these studies.
   - b) The UCF101 and HMDB51 datasets are relatively simple and do not adequately demonstrate the advantages of synthetic data for video pre-training. Experiments on more complex datasets, like Something-SomethingV2, are needed to support the claims.
   - c) Section 6 mentions that this is merely a preliminary work and suggests additional tasks or models will be discussed in future work, leading to the conclusion that this paper does not constitute a complete study.

2. The overall writing resembles a technical (experimental) report, particularly in Section 5, which is structured similarly to [1]. While the paper presents work on videos, it does not adequately expand on this area compared to the image work done by [1].

3. The two synthetic datasets generated, “Accelerating Transforming StyleGAN Crops” and “Accelerating Transforming Textures,” yield good performance for the pre-trained models. However, these datasets are based on data generation techniques introduced by [1].

4. There are writing issues:
   - Page 3, Line 140: Incorrect citation for “Section 3.1.”
   - Page 10, Line 537: The number “92.%” is incomplete.
   - Page 8, Line 424: The origin of “28 datasets” is not explained.

5. Section 5 analyzes how synthetic data benefits video pre-training only from the perspective of static images or individual frames (spatial information). Given that videos are characterized by temporal dynamics and motion cues, the paper lacks relevant experiments and analyses in these areas.

6. The accuracy of the “Dynamic StyleGAN videos” setting in Table 3 is reported to be only 68.7%, but the paper does not provide an explanation or conclusion regarding this result.

[1]	Manel Baradad, Jonas Wulff, Tongzhou Wang, Phillip Isola, and Antonio Torralba. Learning to see by looking at noise. In A. Beygelzimer, Y. Dauphin, P. Liang, and J. Wortman Vaughan (eds.), Advances in Neural Information Processing Systems, 2021. URL https://openreview. net/forum?id=RQUl8gZnN7O.

[2]	YoWhan Kim, Samarth Mishra, SouYoung Jin, Rameswar Panda, Hilde Kuehne, Leonid Karlinsky, Venkatesh Saligrama, Kate Saenko, Aude Oliva, and Rogerio Feris. How transferable are video representations based on synthetic data? In Thirty-sixth Conference on Neural Information Processing Systems Datasets and Benchmarks Track, 2022. URL https://openreview. net/forum?id=lRUCfzs5Hzg.

[3]	Howard Zhong, Samarth Mishra, Donghyun Kim, SouYoung Jin, Rameswar Panda, Hilde Kuehne, Leonid Karlinsky, Venkatesh Saligrama, Aude Oliva, Rogerio Feris. Learning Human Action Recognition Representations Without Real Humans. In Thirty-seventh Conference on Neural Information Processing Systems Track on Datasets and Benchmarks. URL https://openreview.net/forum?id=UBbm5embIB

**Questions:**

1. Could you provide experimental results from other datasets, such as Something-SomethingV2?
2. What type of generator is used in this study? The paper mentions an on-the-fly generation strategy for training (Page 4, Line 215). Does this approach consume more computational resources than the original pre-training? Please provide relevant experimental data.
3. In the paragraph on Page 6, Line 292, how were the experimental data obtained? Is there a missing table or figure? How was the conclusion reached? Additionally, how is the “97.2%” figure mentioned multiple times in the paper derived? How are the experimental data in Section 4.3 calculated?

---

### Note · Authors · 2024-11-13

I have read and agree with the venue's withdrawal policy on behalf of myself and my co-authors.